# ICAFormer: An Image Dehazing Transformer Based on Interactive Channel Attention

**DOI:** 10.3390/s25123750

**Published:** 2025-06-15

**Authors:** Yanfei Chen, Tong Yue, Pei An, Hanyu Hong, Tao Liu, Yangkai Liu, Yihui Zhou

**Affiliations:** 1Hubei Key Laboratory of Optical Information and Pattern Recognition, School of Electrical and Information Engineering, Wuhan Institute of Technology, Wuhan 430205, China; cyf@wit.edu.cn (Y.C.); hhyhong@wit.edu.cn (H.H.); 22303010092@stu.wit.edu.cn (T.L.); 22303010094@stu.wit.edu.cn (Y.L.); 22303010137@stu.wit.edu.cn (Y.Z.); 2School of Artificial Intelligence and Automation, Huazhong University of Science and Technology, Wuhan 430072, China; anpei96@hust.edu.cn

**Keywords:** image dehaze, Transformer, feature extraction, attention mechanism

## Abstract

Single image dehazing is a fundamental task in computer vision, aiming to recover a clear scene from a hazy input image. To address the limitations of traditional dehazing algorithms—particularly in global feature association and local detail preservation—this study proposes a novel Transformer-based dehazing model enhanced by an interactive channel attention mechanism. The proposed architecture adopts a U-shaped encoder–decoder framework, incorporating key components such as a feature extraction module and a feature fusion module based on interactive attention. Specifically, the interactive channel attention mechanism facilitates cross-layer feature interaction, enabling the dynamic fusion of global contextual information and local texture details. The network architecture leverages a multi-scale feature pyramid to extract image information across different dimensions, while an improved cross-channel attention weighting mechanism enhances feature representation in regions with varying haze densities. Extensive experiments conducted on both synthetic and real-world datasets—including the RESIDE benchmark—demonstrate the superior performance of the proposed method. Quantitatively, it achieves PSNR gains of 0.53 dB for indoor scenes and 1.64 dB for outdoor scenes, alongside SSIM improvements of 1.4% and 1.7%, respectively, compared with the second-best performing method. Qualitative assessments further confirm that the proposed model excels in restoring fine structural details in dense haze regions while maintaining high color fidelity. These results validate the effectiveness of the proposed approach in enhancing both perceptual quality and quantitative accuracy in image dehazing tasks.

## 1. Introduction

Haze is a natural atmospheric phenomenon caused by the presence of numerous suspended particles, such as water droplets and dust. Under hazy weather conditions, atmospheric light is absorbed and scattered by these particles during propagation, leading to a gradual attenuation of light intensity before reaching the imaging sensor. This reduction in light intensity significantly impairs the visibility of the surrounding environment, resulting in a degradation of image quality to varying degrees. Images captured under hazy conditions often suffer from diminished clarity, contrast, and color vibrancy. In many practical applications—including public transportation safety, satellite image analysis, outdoor video surveillance, and autonomous driving—high-quality images are crucial for accurate information extraction and decision making. Narasimhan and Nayar et al. [1] formalized the atmospheric scattering model as follows:(1)$$I\left(x\right)=J\left(x\right)t\left(x\right)+A\left(1-t\left(x\right)\right)$$
where *I*(*x*) represents the image affected by haze, *J*(*x*) represents the clear, haze-free image, *A* denotes the global atmospheric light, and *t*(*x*) represents the transmission rate. In the case of a homogeneous atmosphere, the transmission rate can be expressed as follows:(2)$$t\left(x\right)=e^{-\beta d\left(x\right)}$$
where *β* represents the scattering coefficient and *d*(*x*) represents the scene depth. Equation (2) indicates that when *d*(*x*) approaches infinity, *t*(*x*) approaches zero. Combining the two equations above, we have(3)$$A=I\left(x\right),\;d\left(x\right)\to \infty$$

In the practical imaging process of a distant view, *d*(*x*) cannot be infinite, but rather a long distance, which results in a very low transmission rate *t*_0_. Instead of relying on Equation (3) to obtain the global atmospheric light *A*, a more accurate global atmospheric light *A* can be derived using the following equation:(4)$$A=\begin{array}{c} max\\ y\in \left\{x\left|t(x)\leq t_{0}\right.\right\} \end{array}I(y)$$

The above formulation highlights that the core challenge in image dehazing lies in accurately estimating the transmission map. By obtaining a reliable transmission map, the original haze-free image can be effectively restored. However, the precise and direct recovery of the transmission map typically requires additional depth information, which is often unavailable in practical scenarios.

It can be seen that image dehazing is an ill-posed problem. Early image dehazing methods used prior constraints to solve the problem space. They typically estimate A and *t*(*x*) separately to reduce the complexity of the problem, and then use Equation (1) to obtain the result. These prior-based methods can generate images with good visual clarity. However, these images are visually different from clear images, and may introduce artifacts in regions that do not satisfy the prior conditions.

In 2011, He et al. [2] discovered that in a natural image, there is at least one region where the pixel values are very low, which is referred to as the dark channel. Based on this discovery, they proposed the dark channel prior dehazing algorithm. In 2014, Fattal et al. [3] found that in natural images, colors exhibit a one-dimensional distribution within local regions. Building on this, they utilized features with lighter colors and incorporated the smoothness of the transmission map to enhance the image visibility. In 2016, Ren et al. [4] constructed a multi-scale convolutional neural network to model the mapping between hazy image features and transmission rate. This approach significantly improved the scene adaptability of dehazing algorithms. In the same year, Cai et al. [5] designed an end-to-end deep learning model, which directly predicts the transmission map and innovatively couples it with the classic atmospheric scattering model, achieving physically interpretable image restoration. The following year, Li et al. [6] made groundbreaking improvements to the traditional physical model. Through mathematical derivation, they merged the two key parameters, atmospheric light intensity and transmission rate, into a unified variable, creating a simplified parameter model for hazy imaging. While this approach improved the dehazing effect compared with previous methods, the simultaneous estimation of these two parameters introduced cumulative errors in image restoration, which negatively affected the dehazing results. In 2018, Zhang et al. [7] innovatively constructed a dense skip-connected encoder–decoder network with a U-shaped symmetric topology design and proposed a dual-branch collaborative optimization mechanism to jointly estimate the haze concentration parameters. The dehazed image was obtained by solving the atmospheric scattering equation in reverse. The study used a deep learning framework to achieve the end-to-end prediction of transmission rate and atmospheric light parameters in the traditional physical model. In 2021, Zhang et al. [8] improved and optimized the transmission rate, atmospheric light, and dehazed results, and integrated the refined dehazing results with the reconstructed clear image perception, producing more natural and clearer dehazing results. Although dehazing methods based on physical prior constraints have demonstrated significant restoration effects in conventional scenarios after long iterations and evolution, they still experience performance fluctuations under extreme conditions, such as complex lighting and uneven haze distribution, limiting the expansion of their practical application range. The current models still require improvements in their ability to model the nonlinear coupling relationships between atmospheric parameters, which has become a major technical bottleneck limiting the robustness of the algorithms.

In recent years, deep learning has become very popular in the field of computer vision, and researchers have proposed a large number of image dehazing methods based on deep convolutional networks. These methods can be divided into three types: The first type combines the concept of the atmospheric scattering model with deep learning [5,6,7], using deep learning techniques to correct the transmission rate and atmospheric light, and then incorporating them into the model to obtain a clear image. The second type is based on the idea of Generative Adversarial Networks (GANs) [9,10,11], introducing adversarial loss, *L*_1_ loss, and perceptual loss to effectively remove haze from degraded images. The third type utilizes a large dataset of synthetic images [12] as the input to learn image features for dehazing, without the need to estimate the transmission rate and atmospheric light.

Generative Adversarial Networks (GANs), as an innovative unsupervised learning paradigm, have a core architecture consisting of two modules, a generator and a discriminator, forming an adversarial training framework. The model creates a dynamic game mechanism between the generator and the discriminator, where the generator aims to synthesize samples that closely approximate the distribution of real data, while the discriminator continually optimizes its ability to distinguish between real and synthetic data. This adversarial training process drives both networks to reach a Nash equilibrium in the parameter space, ultimately enabling the generative model to accurately model complex data distributions. GANs have achieved great success in image processing. Traditional convolutional neural network algorithms use pixel-level traditional losses, which can lead to blurry output images. GANs introduce high perceptual loss in the feature space, improving the quality of dehazed images. Goodfellow et al. [9] proposed GANs, generating clean images from input images with random noise. However, due to the instability of the training process, synthesized images are easily affected by color shifts and noise. Mirza et al. [10] proposed Conditional GANs (cGAN), which improve the stability of the training process by adding conditional information and variables. Li et al. [13] applied cGAN to single image dehazing, achieving dehazing by introducing *L*_1_ regularization gradients and VGG features. Compared with traditional cGANs, Runde’s proposed encoder–decoder structure is an end-to-end design that generates better results. However, unsupervised GAN-based algorithms also have some drawbacks. First, these GAN algorithms require substantial computational resources and time during the training process. Second, their training is unstable and requires adjusting a large number of parameters. Finally, when applied to dehazing, GANs have limitations in local feature extraction and cannot effectively perceive global feature information of the image.

This study presents a novel single image dehazing model built upon the Transformer architecture. The core design philosophy draws inspiration from the remarkable success of Transformers in the field of natural language processing. Recent advancements have demonstrated that vision Transformers exhibit superior performance across various computer vision tasks, posing a paradigm shift from traditional convolutional neural network (CNN)-based architectures. Guo et al. [14] leveraged Transformer-based architectures to enhance the modeling of long-range dependencies in images, thereby improving dehazing performance. Similarly, Zhao et al. [15] proposed an end-to-end single image dehazing approach that integrates the Transformer architecture with a texture-enhanced attention mechanism, further advancing feature representation capabilities. Specifically, in traditional methods, the feature representation is constructed through hierarchical convolutional operations, which, when compared with the global attention mechanism of Transformers, reveal clear limitations.

## 2. Related Work

### 2.1. Image Dehazing Based on CNNs

In recent years, Convolutional Neural Networks (CNNs) have made significant progress in the field of visual restoration, with image dehazing technology emerging as a particularly prominent application. Existing research primarily follows two major technical approaches: physics-based model-driven methods and end-to-end data-driven methods. The first approach is based on the atmospheric scattering model for constructing a mathematical framework, in which neural networks are employed to estimate the medium transmission rate and atmospheric light parameters. Typical examples include the DehazeNet [5], DCPDN [7], and AOD-Net [6] architectures. These methods perform well in scenes that align with physical assumptions, but their dehazing performance significantly deteriorates when there are deviations in the estimation of atmospheric optical parameters. The second approach adopts an end-to-end mapping strategy, directly regressing a clear image from a hazy input. While this method eliminates the reliance on physical models, it inherently struggles to model long-range pixel dependencies effectively.

### 2.2. Vision Transformer

In 2020, the ViT architecture [16] demonstrated its potential as an alternative to CNNs and achieved significant breakthroughs in the field of computer vision. SwinIR [17] combined the Swin Transformer layers with convolutional layers to form a residual module, which was successfully applied to image restoration tasks. The Swin Transformer [18] divides an image into windows and performs the self-attention mechanism within these windows, ensuring that the computational cost grows linearly. Restormer [19] further improved image restoration by incorporating multi-head transposed attention modules and gated feed-forward networks, enabling high-resolution image restoration. Additionally, ViT has achieved state-of-the-art (SOTA) performance in various tasks, including image enhancement [20,21,22], image segmentation [23,24,25], object detection [26,27,28], human pose estimation [29,30,31], and image classification [32,33,34].

Swin Transformer is one of the pioneers in applying Transformers to low-level vision tasks. It constructs a large residual block consisting of stacked Swin Transformer blocks and convolutional layers. Here, we will briefly review the Swin Transformer. Given a feature map *X*, it projects *X* into *Q*, *K*, and *V* (query, key, and value) through a linear layer, then divides it into different windows, where the partitioning regions of *Q*, *K*, and *V* differ for each window. The calculation formula for the self-attention mechanism is as follows:(5)$$Attention(Q,K,V)=Softmax\left(\frac{QK^{T}}{\sqrt{d}}\right)V$$
where *d* is the dimension of the query and key vectors, used to scale the dot product for numerical stability and to prevent excessively large gradients.

The Swin Transformer adopts a partitioned self-attention mechanism, leveraging a windowing strategy to improve computational efficiency while maintaining the ability to model long-range dependencies. This approach enhances the flexibility of the network while preserving computational efficiency.

### 2.3. Efficient Attention Mechanism

The attention mechanism has been widely adopted in the field of computer vision due to its ability to highlight the relative importance of different features by assigning them adaptive weights. Depending on the focus of the operation, attention mechanisms can be broadly categorized into channel attention, spatial attention, and self-attention, among others. The attention mechanism proposed in this paper is the channel attention mechanism. Compared with other attention mechanisms, the channel attention mechanism is capable of capturing specific patterns or features within an image while minimizing computational complexity, particularly in the task of image dehazing. The most classic implementation of channel attention is the SE (Squeeze-and-Excitation) attention mechanism, proposed by Hu [35] in 2018. Its core idea is to explicitly model the dependencies between channels and adaptively adjust the feature responses of each channel. The formation process of the SE attention mechanism can be divided into three steps: Squeeze, Excitation, and Scale. Let us review the formation process of the SE attention mechanism: Let the input feature map be $U\in R^{H\times W\times C}$, where *H*, *W*, and *C* represent the height, width, and the number of channels, respectively. The spatial dimensions *H* × *W* are compressed by performing global average pooling on each channel *c*.(6)$$z_{c}=\frac{1}{H\times W}\sum_{i=1}^{H}\sum_{j=1}^{W}u_{c}(i,j)$$

Each element *z_c_* represents the global information of the *c*-th channel. The dimensionality of *z_c_* is reduced through a fully connected layer as follows:(7)$$s=W_{2}ReLU(W_{1}\cdot z_{c}+b_{1})+b_{2}$$
where $W_{1},W_{2}\in R^{\frac{C}{r}\times C}$ are the weight matrices, and *b*_1_, *b*_2_ are the biases. After normalizing the weight *s*, it is multiplied with the input feature map ***U*** channel-wise to obtain the weighted feature map $\tilde{U}\in R^{H\times W\times C}$. The channel attention mechanism proposed in this paper similarly enhances the feature representation ability of convolutional neural networks.

## 3. The Proposed Method

Our objective is to develop an efficient and lightweight Transformer-based network for image dehazing. To reduce computational complexity while enhancing the extraction of fine-grained image features, we adopt a Swin Transformer-based framework. Furthermore, we introduce a multi-level convolutional attention mechanism to effectively capture features across different representation levels.

### 3.1. The Overall Architecture of the Network

U-Net [36] is one of the commonly used networks in image processing, capable of capturing multi-scale image features. Inspired by this, the main framework of our model is shown in Figure 1, with a hierarchical design consisting of four stages. The input image $I\in R^{H\times W\times C}$ is received through a 3 × 3 convolution, which extracts local texture features and outputs a feature map $I^{\prime }\in R^{H\times W\times C}$ with the same resolution. Global dependencies are modeled on the feature map to capture long-range spatial relationships. The resolution is halved and the number of channels is doubled through a downsampling operation, resulting in $I^{\prime \prime }\in R^{\frac{H}{2}\times \frac{W}{2}\times 2C}$, where more complex features are modeled in a higher dimensional space. Another downsampling operation is applied, halving the resolution again and doubling the channels, producing $D\in R^{\frac{H}{4}\times \frac{W}{4}\times 4C}$, which enhances the retention of global feature details. The last two stages are the symmetric processes of the first two stages. $D\in R^{\frac{H}{4}\times \frac{W}{4}\times 4C}$ is upsampled to obtain $J^{\prime \prime }\in R^{\frac{H}{2}\times \frac{W}{2}\times 2C}$, and upsampled again to obtain $J^{\prime }\in R^{H\times W\times C}$. We assign weights to the features at each step and fuse the features from different layers to achieve the dehazing effect.

The Transformer block is designed by replacing the standard feed-forward network with a haze-specific feature extraction module, enabling the capture of haze-related features across multiple scales.

Following average pooling and activation via the Sigmoid function, the feature fusion module employs an interactive channel attention mechanism to extract and integrate features across multiple hierarchical levels.

The ICA module is developed based on an interactive channel attention mechanism. We introduce this channel attention mechanism to enhance the representation capability of features at different scales. This design is incorporated into the decoder part of the U-Net architecture (as illustrated in Figure 1), corresponding to the integration of multi-scale features.

Considering that multiple operations within the Transformer block may result in the loss of some image detail information, we incorporate both local and global residual connections. Within the Transformer block, a standard residual connection structure is employed to ensure stable gradient propagation, thus preventing the vanishing gradient problem during Transformer training. Additionally, cross-layer global residual connections are implemented between each basic layer throughout the network. This operation mitigates the vanishing gradient issue in deep networks and facilitates the fusion of features at different scales.

### 3.2. Transformer Block

To effectively enhance feature representation in the single image dehazing task, this paper proposes an efficient visual Transformer block module, as illustrated in Figure 2. This module improves upon the Swin Transformer block by integrating a dual-path design that simultaneously captures long-range dependencies and restores local spatial details.

The input features are normalized through the RescaleNorm layer. Unlike traditional LayerNorm, RescaleNorm preserves the global mean and variance information of the input features, preventing the loss of overall image brightness and contrast due to normalization in low-level vision tasks.(8)$$RescaleNorm(x)=s\cdot \frac{x-\mu }{\sigma (x)}+b$$
where *σ*(*x*) denotes the standard deviation of the input feature *x*, *s* is the learnable scaling coefficient, *μ* is the mean value, and *b* is the bias term. The formula for computing *σ*(*x*) is:(9)$$\sigma (x)=\sqrt{\frac{1}{n}\sum_{i=1}^{n}{\left(x_{i}-\mu \right)}^{2}+\varepsilon }$$
where, as in Equation (8), *μ* is the mean value, and *ε* is a constant introduced for numerical stability.

The features are processed through two distinct paths. The first path utilizes the standard Multi-Head Self-Attention (MHSA) mechanism. The input features are first mapped into query, key, and value vectors, where queries and keys are generated through linear layers. A dot product attention calculation (using MatMul and SoftMax operations) is then performed, followed by the merging and fusion of the attention head outputs through a linear layer to obtain global context information. The second path employs convolutional operations, which complement the local modeling capability of MHSA and enhance the extraction of spatial structural information. This convolutional branch directly operates on the value branch and runs in parallel with the self-attention path, aiding in the recovery of detail and texture information.

To further enhance the model’s feature representation capability, an affine transformation is applied after the fusion of the multi-head attention and convolution branches’ outputs, improving the model’s adaptability to the feature space. The fused features are then fed into a Multi-Layer Perceptron (MLP) module, which consists of two linear layers with a ReLU activation function in between. Compared with the commonly used GELU activation function, ReLU is more suitable for low-level vision tasks, such as image restoration, as it effectively improves the model’s stability and reconstruction quality.

Moreover, the Transformer block incorporates multiple residual connections, which ensure efficient feature transfer between different modules and maintain gradient stability. This design accelerates model convergence during training and enhances the overall dehazing performance.

### 3.3. Interactive Channel Attention Mechanism

In recent years, the SE channel attention mechanism has been widely used in the field of image dehazing, achieving impressive results. The fully connected layers in the SE channel attention mechanism capture global information, enhancing the dehazing network’s ability to focus on key features, thereby improving performance. However, this approach lacks interaction between global and local information, leading to inaccurate feature weight distribution in image dehazing tasks. To efficiently integrate global and local information and achieve a more reasonable weight distribution, we introduce an interactive channel attention mechanism. The framework structure is shown in Figure 3. Our mechanism facilitates bidirectional interaction between local textures and global context during feature fusion, enabling more precise attention weight assignment across hierarchical features.

The interactive channel attention mechanism is primarily utilized during the feature fusion stage of the backbone network. In the process of network construction, effectively leveraging both global and local information is crucial for enhancing the performance of image dehazing.

As shown in Figure 3, given a feature map $X\in R^{H\times W\times C}$, where *C*, *H*, and *W* denote the number of channels, height, and width, respectively. Global information from the channels is first extracted through global average pooling as follows:(10)$$U_{c}=GAP(X_{c})=\frac{1}{H\times W}\sum_{i=1}^{H}\sum_{j=1}^{W}X_{c}(i,j)$$
where *X_c_*(*i*,*j*) is the feature value of the *c*-th channel and *GAP*(*x*) is the global average pooling function. This function allows the feature map to be transformed from *C* × *H* × *W* to *C* × 1 × 1.

*U_c_* represents the statistical measure of the *c*-th channel, while ***U*** denotes the global average pooled feature vector, which can be expressed as ***U*** = [*U*_1_,*U*_2_,…,*U_c_*]^T^.

To extract the global information of the channels, we obtain a portion of the information from the globally average pooled feature ***U***, which primarily focuses on the weight of its own channel without considering the interactions between channels as follows:(11)$$U_{gc}=diag(U)=\sum_{i=1}^{c}U\cdot d_{i}$$
where *diag*(*x*) denotes the diagonalization operation, which is used to extract the global information of the channel features, and *d_i_* represents the global weight vector of the *i*-th channel, which models the overall inter-channel relationships. During the attention computation, it is essential to incorporate dependencies between adjacent channels to enable effective information interaction across channels. Additionally, a portion of the information is extracted from ***U*** as follows:(12)$$U_{lc}=band(U)=\sum_{i=1}^{k}U\cdot b_{i}$$
where *band*(*x*) represents the band-like operation used to extract local channel correlations and *b_i_* represents the *i*-th banded matrix convolution kernel.

In summary, the *diag*(*x*) operation performs channel-wise weighting through a fully connected transformation across channels, modeling global inter-channel dependencies and generating the output *U_gc_*. In contrast, the *band*(*x*) operation applies channel-wise weighting via a localized convolution across channels, capturing local inter-channel dependencies and producing the output *U_lc_*.

Next, matrix multiplication is performed to obtain the interaction information between channels as follows:(13)$$S_{gc}=\sigma \left(\sum_{j}^{c}\left({U_{gc}\cdot U_{lc}}^{T}\right)i,j\right),S_{lc}=\sigma \left(\sum_{j}^{c}\left({U_{lc}\cdot U_{gc}}^{T}\right)i,j\right),i\in 1,2,3\ldots c$$
where *σ*(*x*) represents the activation function applied to *x*, ensuring that the attention weights lie within the range (0, 1), *i* denotes the global channel index corresponding to *U_gc_*, and *j* denotes the local channel index corresponding to *U_lc_*. The operation above sums these two matrices to compute the inductive information for each channel.

Finally, the calculated attention weights *W* are applied to the original input features *X* as follows:(14)$$X^{*}=X\times W$$
where *X* is the input feature map and *X** is the final output feature map with a shape of *C* × 1 × 1. This allows for the weighted adjustment of each channel of the input *F*, enabling the network to focus more on the key channels and enhancing the feature representation capability.

### 3.4. Loss Functions

To effectively enhance the model’s ability to recover fine image details and structural information, this paper introduces a combination of Mean Absolute Error (MAE) loss, perceptual loss, and edge loss. These loss functions guide the model in learning the differences between hazy and clear images. The total loss is defined as a weighted combination of these three components:(15)$$L=\lambda_{1}L_{1}+\lambda_{2}L_{per}+\lambda_{3}L_{\mathrm{edge}}$$
where *λ*_1_, *λ*_2_, and *λ*_3_ represent the respective weighting coefficients assigned to each loss function. Based on the results obtained from controlled experiments, the optimal values were empirically determined to be λ_1_ = 0.9, λ_2_ = 0.05, and λ_3_ = 0.05.

The mean absolute error loss (*L*_1_ loss) function is specified as follows:(16)$$L_{1}=\frac{1}{N}\sum_{i=1}^{N}\left|\hat{I_{i}}-I_{i}\right|$$
where $\hat{I_{i}}$ denotes the model’s predicted value for the *i*-th pixel, *I_i_* represents the corresponding pixel value in the ground-truth clear image, and N indicates the total number of pixels in the image. Compared with *L_2_* loss optimization, the *L_1_* loss effectively reduces oversmoothing artifacts in image dehazing while demonstrating superior detail preservation capability. Furthermore, the inherent insensitivity to outliers in *L*_1_ loss enhances training stability and model robustness.

The *L_per_* loss function is implemented using VGG-16 [37] as the loss network, and its formulation is as follows:(17)$$L_{per}={\sum_{i=1}^{3}\frac{1}{C_{j}H_{j}W_{j}}\left\|\phi_{j}\left(I\right)-\phi \left(J\right)\right\|}_{2}$$
where *C_j_*, *H_j_*, and *W_j_* denote the channel dimension, height, and width of *ϕ_j_*( ), respectively, with *ϕ_j_*(*I*) representing the feature maps of the dehazed image and *ϕ_j_*(*J*) corresponding to the feature maps of the ground-truth clear image.

The *L_edge_* loss function is calculated by applying convolution operations with the Laplacian operator to extract the edge maps of both the clear image and the dehazed image. The loss is then defined as the difference between these edge maps. The formulation is specified as follows:(18)$$L_{edge}=\sum_{i,j}\left|\nabla x_{g}\left(i,j\right)-\nabla x_{r}\left(i,j\right)\right|$$
where ∇*x_g_*(*i*,*j*) denotes the gradient of the generated image at position (*i*,*j*) and ∇*x_r_*(*i*,*j*) represents the gradient of the real image at the corresponding spatial coordinates.

## 4. Experiments and Results

### 4.1. Dataset

The dataset used in this paper is sourced from the RESIDE dataset [38], which consists of multiple subsets for comprehensive evaluation: a synthetic indoor image pair collection (Indoor Training Set, ITS) containing 13,990 samples, an outdoor training subset (Outdoor Training Set, OTS) with 72,135 images, and a Synthetic Objective Testing Set (SOTS) comprising 500 indoor and 500 outdoor hazy images for quantitative assessment. To ensure the fairness and reliability of the experimental results, the dataset was split into a training set and a testing set at a ratio of 9:1.

### 4.2. Evaluation Metrics

The evaluation metrics employed in this study are Peak Signal-to-Noise Ratio (PSNR) and Structural Similarity Index Measure (SSIM). PSNR is an objective metric calculated based on pixel-wise error that is used to quantify the difference between the restored image and the reference image. Essentially, it represents the signal-to-noise ratio derived from a logarithmic transformation of the mean squared error (MSE). A higher PSNR value indicates greater similarity between the restored and reference images, reflecting better image quality. SSIM, on the other hand, is an image quality metric that aligns more closely with human visual perception, comprehensively evaluating the brightness, contrast, and structural information of the image.

Due to the lack of corresponding ground-truth references in real-world images, which are required for the calculation of PSNR and SSIM, we selected FADE [39]—one of the representative methods among numerous no-reference perceptual haze density estimation techniques [40,41]. FADE is capable of predicting haze density directly from a single hazy image without the need for a corresponding haze-free reference. The haze density is computed as follows:(19)$$FADE=\frac{D_{f}}{D_{ff}+1}$$
where *D_f_* denotes the Mahalanobis distance between the input image and the statistical model of haze-free images, while *D_ff_* represents the Mahalanobis distance between the input image and the statistical model of hazy images. The Mahalanobis distance is calculated as follows:(20)$$D=\frac{1}{N}\sum_{i=1}^{N}\sqrt{{\left(x_{i}-\mu \right)}^{T}\sum^{-1}\left(x_{i}-\mu \right)}$$
where *μ* denotes the mean feature vector of haze-free or hazy images, Σ represents the corresponding covariance matrix of the feature vectors for haze-free or hazy images, and *x_i_* denotes the 12-dimensional feature vector extracted from the *i*-th image patch. The higher the FADE score, the more the image resembles a hazy image.

### 4.3. Comparison Experiment of Synthetic Dataset

The data presented in Table 1 demonstrate that the traditional Dark Channel Prior (DCP) algorithm underperforms in dehazing metrics, with its PSNR (18.7 dB) and SSIM (0.76) significantly below the benchmark values of 20 dB and 0.8, respectively. In contrast, classical deep learning-based models like DehazeNet and AOD-Net have achieved substantial improvements in both metrics. Notably, the MSBDN model enhanced PSNR to 29.73 dB/27.75 dB and SSIM to 0.954/0.935 through architectural optimizations. Of particular significance, the proposed method in this study achieves substantial improvements of 0.53 dB (indoor) and 1.64 dB (outdoor) in PSNR, along with 1.4% and 1.7% gains in SSIM, respectively, while maintaining model compactness. These results validate that our novel dehazing network more accurately reconstructs textural details and color distribution in haze-free images compared with MSBDN [42].

The performance comparison between the proposed algorithm and existing methods on the synthetic SOTS dataset is illustrated in Figure 4. The first three experimental groups showcase the dehazing results in indoor scenarios, while the following four groups present the outcomes in outdoor environments. The experimental analysis indicates that images processed by the DCP algorithm suffer from widespread brightness attenuation and are prone to color distortion, particularly in the high-light regions of the original hazy images. Although DehazeNet effectively removes haze, residual haze remains visible in some samples, accompanied by noticeable chromatic aberrations. AOD-Net improves upon these methods but tends to cause overexposure in darker regions. Among the MSBDN [42], FSAD-Net [43], and PAN [44] algorithms, MSBDN demonstrates superior performance in both subjective and objective evaluations. While MSBDN achieves better dehazing performance, comparative analysis with our proposed algorithm and haze-free ground-truth images reveals its limitations in restoring color in local regions. Both subjective and objective evaluations confirm that the proposed algorithm outperforms the comparative methods in terms of visual quality and quantitative metrics, providing restoration results that more closely resemble real-world scenes. Notably, our method demonstrates exceptional performance in preserving color naturalness.

To emphasize the superiority of the proposed algorithm in dehazing performance, Figure 5 uses a local magnification comparative approach for validation. High-resolution image comparisons reveal significant discrepancies at the microscopic level in traditional methods, with the DCP algorithm showing structural distortions, as specifically illustrated in the magnified regions. Although DehazeNet and AOD-Net effectively remove haze to some extent, their generated images still exhibit insufficient color saturation compared with haze-free ground-truth samples, particularly in high-density haze regions where haze interference is not fully eliminated. In contrast, the proposed algorithm demonstrates remarkable advantages in restoring texture details while maintaining chromatic consistency. It not only effectively removes haze but also preserves visual fidelity in complex scenarios.

### 4.4. Comparison Experiment of Real Dataset

To systematically validate the effectiveness of the proposed method, a multi-dimensional comparative experimental framework was constructed under real-world hazy conditions. As illustrated in Figure 6, the experimental results highlight the key performance limitations of existing algorithms. The DCP algorithm exhibits noticeable deficiencies in color fidelity, including pronounced color gamut compression in sky and road regions (Figure 6b,c), along with residual haze along edge contours (Figure 6a). DehazeNet demonstrates partial dehazing effectiveness under light-haze conditions but fails to fully remove haze in dense regions (Figure 6b). Although AOD-Net shows generally strong performance, it introduces localized artifacts in areas with complex road textures (Figure 6c). The MSBDN algorithm struggles to handle non-uniform haze distributions, resulting in abnormal haze concentration gradients, as observed in Figure 6a,d. The FSAD-Net algorithm tends to produce excessive hierarchical differences, leading to overexposure in regions with significant color gamut variations (Figure 6a,d). The PAN algorithm, on the other hand, fails to achieve satisfactory dehazing performance in the edge regions of the image (Figure 6a,b). Subjective evaluations further confirm that the proposed method outperforms both traditional approaches and state-of-the-art deep learning models in critical aspects such as detail preservation and chromatic consistency, demonstrating its practical superiority in real-world scenarios.

To better perform quantitative evaluation on real-world images, we introduced the FADE algorithm. A lower FADE evaluation value indicates that the image is closer to a haze-free image. As shown in Table 2, our method achieves the best performance on Figure 6a,b,d, and the second-best performance on Figure 6c. This further confirms the superior performance of our algorithm in practical applications.

### 4.5. Comparative Experiment on Params and FLOPs

To assess the practical applicability of the proposed algorithm, we conducted a comparative analysis of the parameter count and floating point operations (FLOPs) across various dehazing models, as summarized in Table 3. Among the existing models, DehazeNet and AOD-Net demonstrate lower computational complexity due to their relatively small parameter and FLOP counts. However, this simplicity may compromise their ability to effectively handle haze with varying densities. On the other hand, models like MSBDN and FSAD-Net involve significantly higher parameter counts and FLOPs, which can limit their deployment in resource-constrained scenarios. Although the PAN algorithm exhibits a similar computational footprint to our method, its dehazing performance remains suboptimal. In contrast, the proposed model achieves a more favorable balance between restoration quality and computational efficiency, underscoring its practical advantages for real-world applications.

### 4.6. Comparative Experiment on Attention Mechanisms

To evaluate the optimization efficacy of the interactive attention mechanism, this study established a comparative analysis framework based on the SOTS dataset. In a controlled variable experimental design, four representative attention modules—SE [35], ECA [45], CBAM [46], and SK [47]—were embedded into a unified network architecture for benchmark testing. The quantitative evaluation data in Table 4 demonstrate that the ICA module achieves a peak structural similarity index (SSIM) of 0.968 and a suboptimal peak signal-to-noise ratio (PSNR) of 30.26 dB. Although ICA yields a slightly lower PSNR compared with SK, it offers greater flexibility for subsequent image processing tasks. Compared with mainstream attention mechanisms, the ICA mechanism exhibits significant advantages in haze-line estimation accuracy and detail reconstruction capability through its multi-scale feature interaction strategy.

## 5. Conclusions

This paper presents ICAFormer, a novel Transformer-based dehazing framework that integrates an interactive channel attention (ICA) mechanism to address the challenges of feature correlation in single image dehazing. By introducing a bidirectional cross-layer attention structure, the model effectively balances global context modeling with local texture enhancement. Additionally, the proposed multi-scale feature fusion strategy further strengthens the network’s ability to preserve fine details and restore color fidelity in dense haze regions.

Extensive experiments on both synthetic and real-world datasets demonstrate the effectiveness of our approach, showing notable improvements over existing state-of-the-art methods in terms of visual quality and quantitative metrics. Our method particularly excels in challenging conditions where detail preservation and contrast recovery are critical.

While the model achieves strong performance, we also acknowledge the remaining challenges under extreme conditions such as highly non-uniform haze or complex lighting. These limitations provide valuable directions for future research, which we aim to explore through lightweight architectures and video-based extensions as outlined in the next section.

## 6. Future Work

In future work, we plan to integrate attention mechanisms from Transformer architectures into more lightweight models to enhance the applicability of the proposed framework. We aim to investigate its implementation in real-time inference for camera-captured images and further explore its extension to video dehazing tasks, thereby providing robust technical support for practical deployment. This direction will emphasize optimizing computational efficiency while maintaining restoration fidelity, particularly addressing temporal consistency and dynamic scene adaptation in video-based applications.

Moreover, we recognize that extreme dehazing scenarios—such as those involving non-uniform haze distribution, complex illumination conditions, or dense occlusions—remain challenging. To this end, we intend to explore improved domain adaptation techniques, robustness-oriented training strategies, and hybrid modeling approaches that combine physical priors with deep features. These advancements are expected to improve generalization in uncontrolled real-world environments and further enhance the reliability of the model under diverse operational conditions.

## Figures and Tables

**Figure 1 sensors-25-03750-f001:**
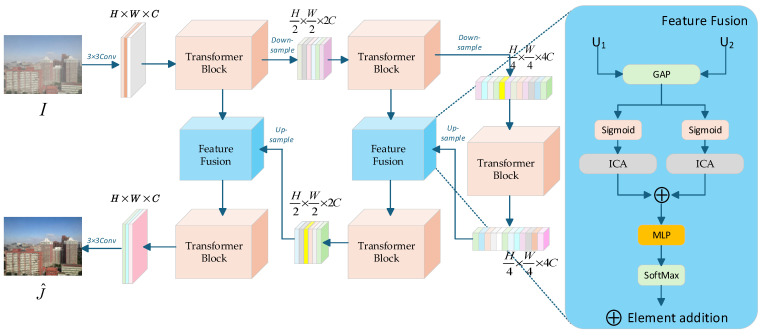
The overall architecture of the network.

**Figure 2 sensors-25-03750-f002:**
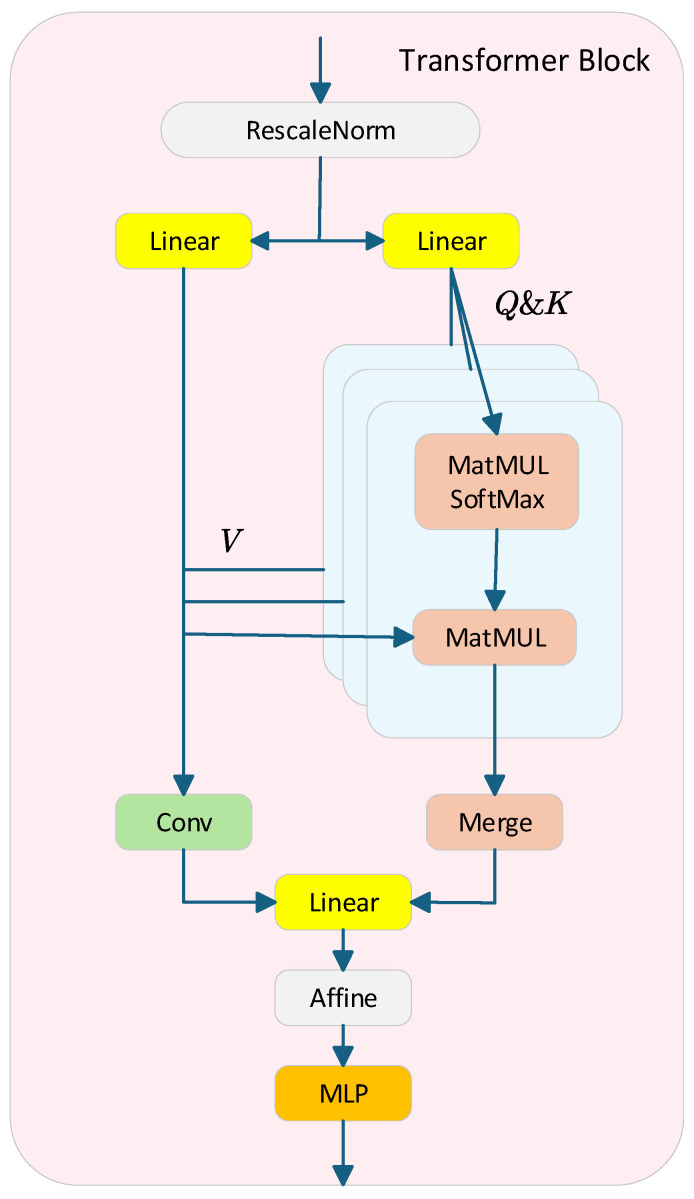
Schematic diagram of Transformer block structure.

**Figure 3 sensors-25-03750-f003:**
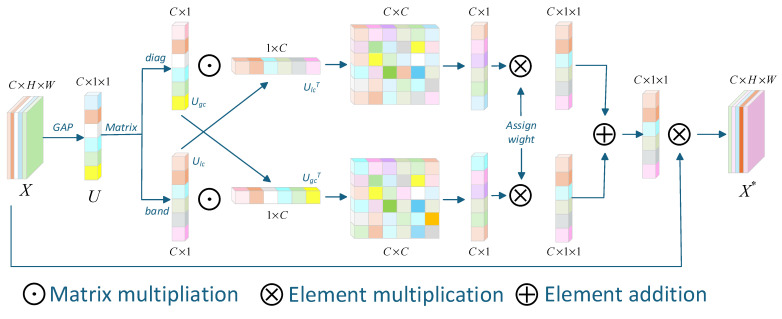
Interactive attention mechanism.

**Figure 4 sensors-25-03750-f004:**
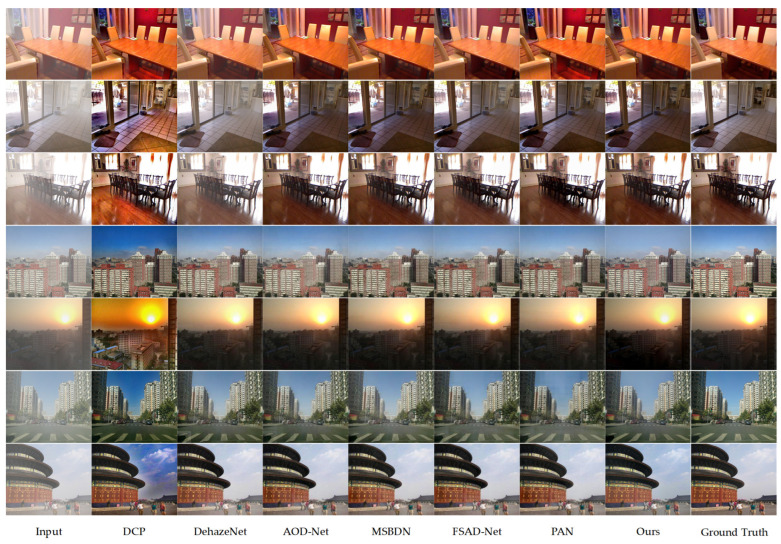
Comparison chart of indoor dehazing methods using ITS (first three sets of photos). Comparison chart of outdoor dehazing methods using OTS (last three sets of photos). From left to right are the original image, DCP, DehazeNet, AOD-Net, MSBDN, FSAD-Net, PAN, our algorithm, and ground truth.

**Figure 5 sensors-25-03750-f005:**
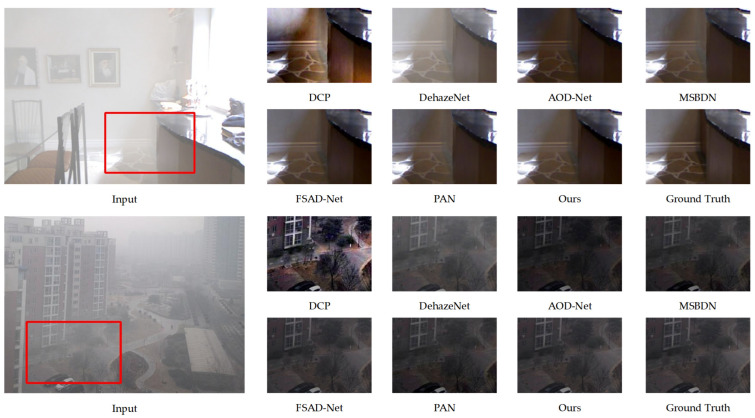
Partial zoom in comparison chart of dehazing methods based on SOTS.

**Figure 6 sensors-25-03750-f006:**
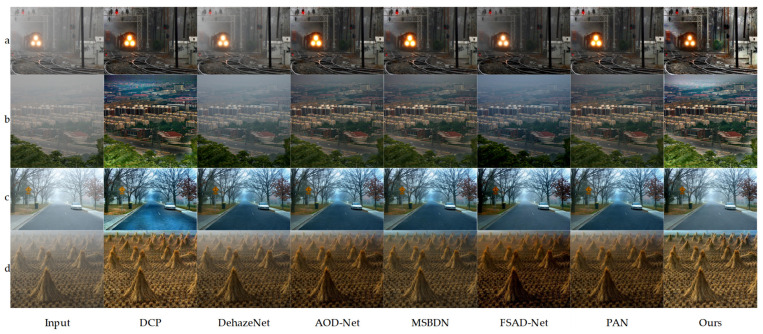
Comparison chart of real-world dehazing methods. From top to bottom are (**a**–**d**). From left to right are the original image, DCP, DehazeNet, AOD-Net, MSBDN, FSAD-Net, PAN, and our algorithm.

**Table 1 sensors-25-03750-t001:** Quantitative comparison of various algorithms on RESIDE SOTS testing set.

Methods	Indoor		Outdoor	
	PSNR	SSIM	PSNR	SSIM
DCP	18.72	0.769	19.13	0.774
DehazeNet	22.75	0.886	21.29	0.877
AOD-Net	24.51	0.919	23.75	0.903
MSBDN	29.73	0.954	27.75	0.935
FSAD-Net	23.41	0.937	26.39	0.929
PAN	23.92	0.923	25.39	0.917
Ours	30.26	0.968	29.39	0.952

**Table 2 sensors-25-03750-t002:** Quantitative comparison of various algorithms on the FADE metric.

Methods	Figure 6a	Figure 6b	Figure 6c	Figure 6d
Input	1.1980	1.2594	0.9634	1.0366
DCP	0.1890	0.1780	0.1788	0.1800
DehazeNet	0.3237	0.5798	0.4277	0.3296
AOD-Net	0.1890	0.2657	0.2320	0.2332
MSBDN	0.2035	0.2807	0.2195	0.2762
FSAD-Net	0.2274	0.2606	0.1934	0.1875
PAN	0.2675	0.3264	0.2076	0.3417
Ours	0.1171	0.1477	0.1830	0.1566

**Table 3 sensors-25-03750-t003:** Comparison of Params and FLOPs for various algorithms on RESIDE SOTS testing set.

Methods	Params/×10^6^	FLOPs/×10^9^
DCP	-	-
DehazeNet	0.009	0.581
AOD-Net	0.002	0.115
MSBDN	31.35	41.54
FSAD-Net	11.27	50.46
PAN	2.611	52.20
Ours	2.517	25.79

**Table 4 sensors-25-03750-t004:** Comparison of various attention mechanisms on RESIDE SOTS testing set.

Attentions	PSNR	SSIM
SE	23.42	0.906
ECA	22.16	0.914
CBAM	25.88	0.931
SK	30.54	0.962
ICA	30.26	0.968

## Data Availability

The RESIDE dataset is made publicly available for research purposes. For more information, please refer to the website: https://sites.google.com/site/boyilics/website-builder/reside/ (accessed on 10 April 2025).

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
