# Peer review of "ICAFormer: An Image Dehazing Transformer Based on Interactive Channel Attention"

_sensors, 2025, doi:10.3390/s25123750_

Round 1
Reviewer 1 Report
Comments and Suggestions for Authors
This paper presents a Transformer-based dehazing model to address the limitations of traditional methods in capturing global dependencies and preserving local details. By integrating a multi-scale feature pyramid and an enhanced cross-channel attention weighting strategy, the network effectively fuses global contextual information with local texture features. Though the architecture of the network is relatively innovative, there remain some problems to be fixed.
- The authors employ several loss functions during training, including mean absolute error loss, perceptual loss, and edge loss. The weights assigned to each loss function should be explicitly provided in the manuscript.
- Although the authors mention the use of the RTTS dataset for experiments in Section 4.1, the manuscript does not provide the corresponding experimental results. The authors should provide and discuss the experimental results on the RTTS dataset.
- To comprehensively evaluate dehazing performance on real-world images, it is recommended to incorporate no-reference dehazing metrics, such as “Dual-stream complex-valued convolutional network for authentic dehazed image quality assessment, IEEE TIP, 10.1109/TIP.2023.3343029”, “Visibility and Distortion Measurement for No-Reference Dehazed Image Quality Assessment via Complex Contourlet Transform, IEEE TMM, 10.1109/TMM.2022.3168438”, “Referenceless Prediction of Perceptual Fog Density and Perceptual Image Defogging, IEEE TIP, 10.1109/TIP.2015.2456502” etc. Citing these no-reference dehazing metrics are necessary.
- The dehazing methods compared in the paper are only up to 2020. Please include more recent methods to ensure a comprehensive and up-to-date comparison.
Language and logic need to be improved. Some sentences and paragraphs fail to express clearly. A native English editor is recommended to help you.
Reviewer 2 Report
Comments and Suggestions for Authors
This is a work that focuses on image dehazing using the Transformer architecture awith interactive channel attention. The authors propose a novel ICAFormer model, which combines the Transformer architecture with an interactive channel attention mechanism for single-image dehazing tasks. Overall, the work has a clear structure, innovative technical methods, and convincing experimental results. However, there are still some aspects that can be further improved.
- In Section 3.4, the description of the perceptual loss is incomplete, and the pre-trained feature extractor being used is not clearly indicated.
- The mathematical derivation of the Interactive Channel Attention (ICA) mechanism lacks sufficient intuitive explanation, making it difficult to understand why the band operation can effectively extract the local correlations among channels.
- In Equation (8) on line 315, the definition of RescaleNorm is incomplete, and the calculation method of σ(x) is not clearly stated.
- The description of the comparative experiment on the attention mechanism is inconsistent with the results in Table 3. Table 3 shows that the SK mechanism outperforms ICA in terms of PSNR, but the text emphasizes the advantages of ICA.
- Conduct additional ablation experiments to analyze the contributions of different components, such as the Interactive Channel Attention (ICA) and Transformer Block, to the final performance.
- In Section 4.4, incorporate a quantitative assessment of the results on real - world datasets.
Reviewer 3 Report
Comments and Suggestions for Authors
Image dehazing is an extremely popular and rapidly evolving topic in computer vision, with a growing number of publications addressing this issue. The paper under review proposes a dehazing method based on a visual transformer with an interactive channel attention mechanism. The article is relevant to the journal’s scope, well-structured, and includes all necessary sections for presenting scientific research. The proposed method appears promising, as confirmed by the experimental results, since it avoids two major challenges in dehazing: atmospheric light estimation and haze homogeneity assumption.
However, there are two main concerns regarding the work:
Neither in the literature review nor in the algorithm comparison are other visual transformer-based models mentioned, despite the fact that such models often use similar architectures and have been known at least since 2022 (e.g. [ Guo, C. L., Yan, Q., Anwar, S., Cong, R., Ren, W., & Li, C. (2022). Image dehazing transformer with transmission-aware 3D position embedding . In Proceedings of the IEEE/CVF Conference on Computer Vision and Pattern Recognition (pp. 5812–5820)]). Furthermore, the significance of the experimental evaluation could be strengthened by including more recent state-of-the-art methods in the comparison.
Unfortunately, the paper does not describe how the data were split into training and testing sets. Based on the current description, it appears that the same data used for training were also used for testing. If this is indeed the case, then the model proposed in the paper has a prior advantage compared to others, which may have been trained on different datasets. In such a scenario, it is hard to determine whether any performance advantage of the proposed approach comes from the data used or from the proposed method itself.
Additionally, there are several inconsistencies in mathematical notation:
In Equations (3) (line 54) and (4) (line 59), the left-hand side uses the symbol α, while the atmospheric light is denoted as 'A' elsewhere.
In Equation (5) (line 180), the symbol "d" is used without explanation, although earlier "d" was introduced as a function d(x) representing distance, potentially leading to confusion.
The original formulation of RescaleNorm [Song, Y., He, Z., Qian, H., & Du, X. (2023)] includes a term for mathematical expectation (mean), which is absent in Equation (8) (line 263) of the manuscript. Please provide clarification regarding this modification.
Overall, the work is sufficiently promising and can be considered for publication after addressing these comments.
Reviewer 4 Report
Comments and Suggestions for Authors
This paper presents the ICAFormer model for solving the problem of image dehazing. The authors propose a Transformer-based dehazing framework integrated with an interactive channel attention mechanism. By building a bidirectional attention mechanism with cross-layer interaction capabilities, a dynamic balance between global context modeling and local texture enhancement is achieved. The authors demonstrate significant performance benefits on both standard datasets and real-world scenarios, successfully preserving details and restoring colors in dense haze regions. The text of the paper is excellently written. The authors provide formulas and precise diagrams, and illustrate the results in detail. The literature references and descriptions of analogs of their approach are also adequate. I have no comments on this. The issue of accessibility of the algorithm remained unclear to me. Will readers be able to use this model for their data?
Round 2
Reviewer 1 Report
Comments and Suggestions for Authors
In the References, I don't find this reference "Referenceless Prediction of Perceptual Fog Density and Perceptual Image Defogging” (IEEE TIP, 2015)".
Moreover, I think the references “Dual-stream complex-valued convolutional network for authentic dehazed image quality assessment, IEEE TIP, 10.1109/TIP.2023.3343029”, “Visibility and Distortion Measurement for No-Reference Dehazed Image Quality Assessment via Complex Contourlet Transform, IEEE TMM, 10.1109/TMM.2022.3168438”, should be cited.
Reviewer 2 Report
Comments and Suggestions for Authors
The revised manuscript has made necessary modifications according to the feedback. The quality has been improved, according to the current state.
Reviewer 3 Report
Comments and Suggestions for Authors
The authors have made significant improvements to the paper and addressed the main issues that were previously noted.
